# Antiprotozoal Aminosteroid Alkaloids from *Buxus obtusifolia* (Mildbr.) Hutch.

**DOI:** 10.3390/molecules30234558

**Published:** 2025-11-26

**Authors:** Justus Wambua Mukavi, Monica Cal, Marcel Kaiser, Pascal Mäser, Njogu M. Kimani, Leonidah Kerubo Omosa, Thomas J. Schmidt

**Affiliations:** 1University of Münster, Institute of Pharmaceutical Biology and Phytochemistry (IPBP), PharmaCampus, Corrensstraße 48, D-48149 Münster, Germany; jmukavi@uni-muenster.de; 2Swiss Tropical and Public Health Institute (Swiss TPH), Kreuzstrasse 2, CH-4123 Allschwil, Switzerland; monica.cal@swisstph.ch (M.C.); marcel.kaiser@swisstph.ch (M.K.); pascal.maeser@swisstph.ch (P.M.); 3University of Basel, Petersplatz 1, CH-4003 Basel, Switzerland; 4Department of Physical Sciences, University of Embu, Embu P.O. Box 6-60100, Kenya; mark.njogu@gmail.com; 5Department of Chemistry, Faculty of Science & Technology, University of Nairobi, Nairobi P.O. Box 30197-00100, Kenya; lkerubo@uonbi.ac.ke

**Keywords:** *Buxus obtusifolia* (Mildbr.) Hutch., Buxaceae, aminosteroid alkaloids, *Trypanosoma brucei rhodesiense*, *Plasmodium falciparum*, natural products

## Abstract

Human African Trypanosomiasis (HAT; sleeping sickness) and Malaria are life-threatening protozoan infections in tropical regions, with limited treatment options. As part of our ongoing efforts to discover new aminosteroid alkaloids from the Buxaceae family with antiprotozoal activity, which might serve as leads to new drugs against these infections, we investigated the dichloromethane extract from the leaves of *Buxus obtusifolia* (Mildbr.) Hutch. collected in Kenya, a species native to Kenya and Tanzania. To the best of our knowledge, and based on the most recent comprehensive literature review, this study represents the first phytochemical investigation of this plant. The alkaloid-enriched fraction yielded a total of 24 aminosteroid alkaloids, including 18 hitherto undescribed compounds (**2**, **3**, **5**–**9**, **11**, **12**, **15**–**19**, and **21**–**24**), along with six known compounds, two of which (**1** and **4**) are described as constituents of a natural source for the first time. Obtusiaminocyclin (**24**) represents the first Buxus alkaloid with a novel carbocyclic steroid skeleton with a cyclopropane ring comprising C-9, C-19 and C-11 accompanied by an unprecedented amino bridge between C-3 and C-10. The structures of the isolated compounds were determined using UHPLC/+ESI-QqTOF-MS/MS and NMR spectroscopy. The total crude extract, the alkaloid-enriched fraction, CPC subfractions and all isolated compounds were tested for in vitro antiprotozoal activity against *Trypanosoma brucei rhodesiense* (*Tbr*, responsible for East African HAT) and *Plasmodium falciparum* (*Pf*, responsible for tropical Malaria) as well as cytotoxicity against mammalian cells (L6 cell line). Deoxycyclovirobuxeine-B (**12**) (IC_50_ = 0.8 µmol/L, SI = 108) and 29-trimethoxybenzoyloxy-obtusibuxoline (**5**) (IC_50_ = 0.5 µmol/L, SI = 11) showed the highest activities with good selectivity indices against *Tbr* and *Pf*, respectively. Consequently, our findings provide valuable aminosteroid candidates that can serve as promising leads in our ongoing search for new drugs against HAT and Malaria.

## 1. Introduction

Human African Trypanosomiasis (HAT), also known as sleeping sickness, and Malaria are vector-borne protozoan infections that affect millions of people, particularly in sub-Saharan Africa and other developing nations [1,2].

HAT is caused by two genetically different protozoan parasites that are transmitted to humans by the bites of infected tsetse flies (*Glossina* spp.); *Trypanosoma brucei gambiense* (*Tbg*), which causes the chronic form in West and Central Africa, currently accounts for around 92% of recorded cases, and *T. b. rhodesiense* (*Tbr*) causes acute illness in East and Southern Africa [3]. While the number of reported cases has steadily decreased since 2010, in 2024 the World Health Organization (WHO) reported a total of only 583 cases of HAT. However, this figure is likely underestimated due to reduced surveillance and lack of reported data in some endemic regions [3]. Although worldwide efforts to manage this disease have greatly decreased its occurrence, underdiagnosis and intermittent outbreaks still pose problems. While eradication of HAT as a public health issue seems within reach, continued research into new chemotherapeutic agents is still crucial, especially in light of the limitations of existing therapies, which are often toxic, costly, complex to administer, or restricted by the disease stage [4,5,6].

Malaria is caused by various species of *Plasmodium*, with *P. falciparum* (*Pf*) being responsible for the most devastating and deadly infections. Infected female *Anopheles* mosquitoes transmit the protozoan parasite into the human body, where the parasite first infects the liver and then multiplies in red blood cells, resulting in a number of symptoms, including organ failure or death [5,7]. Based on a 2024 World Health Organization report, Malaria infections increased by 11 million in 2023, reaching an estimated 263 million cases and 597,000 deaths. More than 95% of both cases and fatalities occurred in the WHO African Region [5]. Despite the availability of artemisinin-based combination therapies (ACTs) and recent advances such as Malaria vaccines, the emergence of artemisinin resistance and decreased sensitivity to partner drugs highlight the urgent need for novel antimalarial compounds with new mechanisms of action [5,8,9].

In the previous studies of our group on steroidal alkaloids from the aerial parts of the European *Buxus sempervirens*, several constituents were identified, some of which exhibited significant activity against *Pf* and *Tbr* [10,11,12,13]. In an initial study, Althaus et al. isolated *O*-tigloylcyclovirobuxeine-B from the dichloromethane leaf extract of *B. sempervirens* as the principal compound responsible for the extract’s antiplasmodial activity [10]. In a more detailed follow-up study by Szabó et al., 25 alkaloids were systematically isolated from the aerial parts of this plant and assessed for antiprotozoal activity. Several compounds exhibited potent activity against *Pf* and *Tbr*, while others showed moderate to low activity [12]. Very recently, aminosteroid alkaloids isolated from *Pachysandra terminalis*, another member of the Buxaceae family studied by Schäfer et al. of our group, demonstrated notable activity against both parasites, further expanding the scope of bioactive scaffolds within the family [14]. These findings underscore the importance of isolating and evaluating further *Buxus* alkaloids for their antiprotozoal potential.

In this context, the present study investigated the antiprotozoal potential of amino-nortriterpenoid alkaloids (termed aminosteroids in this article for simplicity) isolated from *Buxus obtusifolia*, an evergreen shrub whose root decoction is used in the folk medicine of the Giriama community in Kenya against chest complaints [15]. The species is currently classified as vulnerable, highlighting the importance of its conservation. A comprehensive literature review revealed no prior pharmacological or phytochemical investigations on this species, but as a close relative to *Buxus sempervirens*, it promises to contain further potent antiprotozoal alkaloids. Through targeted extraction, fractionation, and structural analysis, this study focused on identifying novel compounds with in vitro activity against *Pf* and *Tbr*. This work further broadens the structural diversity of antiprotozoal aminosteroid alkaloids and contributes valuable candidates to our ongoing search for new therapeutic hits or leads against Malaria and HAT.

## 2. Results and Discussion

### 2.1. In Vitro Antiprotozoal Activity of the Crude Extracts and Fractions

To determine the optimal plant part and extraction solvent, 5 g of the air-dried twigs and leaves were separately extracted on a small-scale using dichloromethane (CH_2_Cl_2_) and methanol (CH_3_OH). The crude extracts were evaluated for their in vitro efficacy against *Trypanosoma brucei rhodesiense* (*Tbr*), and *Plasmodium falciparum* (*Pf*) as well as their cytotoxicity against L6 rat skeletal myoblasts (Cytotox. L6). The results (Table 1) revealed that the dichloromethane extract of the leaves, had the strongest activity and selectivity index (SI) against *Pf* (IC_50_ = 1.1 ± 0.11 µg/mL, SI = 24). These findings are consistent with earlier reports on *Buxus sempervirens*, which showed that leaf extracts exhibit significant and selective in vitro activity against *Plasmodium falciparum* [11,16]. Based on these first biological findings, the CH_2_Cl_2_ extract of the leaves was chosen for further fractionation and isolation of active constituents. For the large-scale extraction, about 680 g of the powdered plant material were exhaustively extracted with CH_2_Cl_2_ in a Soxhlet apparatus set up. In order to separate the alkaloids and lipophilic compounds from the crude extract, an acid-base extraction was carried out. The resulting alkaloid fraction as well as the lipophilic and hydrophilic residuals were subsequently evaluated for antiprotozoal and cytotoxic activities. As indicated in Table 1, the alkaloid fraction demonstrated enhanced antiprotozoal potency against both parasites compared to the previously tested crude extract. Additionally, the alkaloidal fraction showed stronger activity against *Tbr* and *Pf* relative to the lipophilic fraction as expected, whereas the hydrophilic residue was inactive. These findings point to the alkaloids as the main contributors to the antiprotozoal effects of the CH_2_Cl_2_ leaf extract. Consequently, the alkaloid fraction was further fractionated using centrifugal partition chromatography (CPC), followed by isolation of pure compounds.

### 2.2. Isolation and Characterization of Aminosteroids from Buxus Obtusifolia

The fractionation and isolation procedure is summarized in Figure 1. A portion of the alkaloid fraction (5 g) was subjected to CPC fractionation using a biphasic solvent system composed of iso-hexane/ethyl acetate (8/2; *v*/*v*, upper phase) and CH_3_OH/H_2_O/propan-2-ol (8/2/1; *v*/*v*/*v*, lower phase). This process yielded 16 subfractions, which were monitored by TLC and UHPLC/+ESI-QqTOF-MS/MS (henceforth abbreviated LC/MS) and subsequently evaluated for their in vitro efficacy against *Tbr* and *Pf*, as well as for cytotoxicity (Table 2). Fractions 3 through 16 exhibited promising antiplasmodial activity, with IC_50_ values ranging from 0.9 ± 0.1 to 1.9 ± 0.5 μg/mL. Among these, fraction 15 was the most potent. Furthermore, subfractions 3 and 4 demonstrated significant activity against *Tbr* (IC_50_ < 1 μg/mL) with greater selectivity indices (SI = 34 and 21, respectively) compared to the parent alkaloidal fraction. The antiprotozoal activity observed in the CPC subfractions was consistent with previous findings from our group [13], which showed that most CPC subfractions derived from the leaf CH_2_Cl_2_ extract of *B. sempervirens* exhibited greater potency against *Pf* than against *Tbr*. As similarly observed in that study [13], the majority of CPC subfractions in the present work showed a clear preference for inhibiting *Pf* compared to mammalian cytotoxicity (all SI > 10).

Figure 1 provides a summary of the isolation of 24 pure aminosteroid alkaloids from various CPC subfractions using preparative HPLC. Structural elucidation and identification of the isolated compounds (see Figure 1) were accomplished through LC/MS analysis and comprehensive NMR spectroscopic techniques. A total of 15 compounds (**1–15**) with a 9β,10β-cyclo-5α-pregnane skeleton, and seven compounds (**17–23**) featuring the 9 (10→19) *abeo*-5α-pregnane framework, both structural motifs typical of *Buxus* alkaloids, were successfully isolated. In addition to these compounds, an unprecedented C-4 demethylated aminosteroid (**16**) and a novel *Buxus* alkaloid featuring a previously unknown carbocyclic core structure (**24**) were isolated and their structures fully elucidated. Since *Buxus obtusifolia* had not been previously subjected to chemical investigation, all compounds reported herein are described for the first time as constituents of this species. Compounds **2**, **3**, **5–9**, **11**, **12**, **15–19**, and **21–24** are newly identified and have not been previously described in the literature. Compounds **1** and **4**, which were formally reported only as hydrolysis products of *N*-benzoylcycloprotobuxoline-C isolated from *B. sempervirens* [17], are herein described for the first time as natural products. Given the limited spectroscopic data available in the literature for compound **1**, its complete NMR assignments are presented in Table 3 and Table 4. The identification of known compounds cycloprotobuxoline-C (**1**) [17], cycloprotobuxoline-D (**4**) [17], cyclonataminol (**10**) [18], cyclovirobuxeine-A (**13**) [19], cyclovirobuxeine-B (**14**) [20,21], and *O^2^*-natafuranamine (**20**) [18] was accomplished through their high resolution mass spectrometry in combination with 1D- and 2D-NMR spectroscopic analyses, which were in full agreement with previously published data.

Compound **2** was isolated as a yellow gum and its molecular formula was determined by its LC/MS quasimolecular ions at *m*/*z* 217.1949 [M + 2H]^2+^ and 433.3773 [M + H]^+^, as C_27_H_48_N_2_O_2_, which differed from that of the known compound **1** (cycloprotobuxoline-C) by containing an additional oxygen atom (Appendix A). The ^1^H- and ^13^C-NMR data of compound **2** (Table 3 and Table 4, respectively; spectra shown in Appendix A) were generally very similar to those of compound **1**, with notable differences confined to the side chain region. For instance, in the ^13^C-NMR spectrum of compound **2**, δ_C20_ was shifted downfield from 67.6 ppm to 82.8 ppm, δ_C21_ from 11.5 ppm to 16.0 ppm, and δ_C33/34_ from 43.6/35.9 ppm to 57.1/51.8 ppm. These localized downfield shifts suggested *N*-oxidation of the tertiary amine [22,23]. Compound **2** was, therefore, elucidated as cycloprotobuxoline-C *N*_20_-oxide. Notably, this compound was observed in the LC/MS chromatogram of the crude extract, confirming that it is a genuine natural metabolite rather than an artifact generated during isolation. The occurrence of this compound alongside its non-oxidized analog **1** suggests the possibility of an oxidative branch in the *Buxus* alkaloid biosynthetic pathway. However, *N*-oxides are uncommon in *Buxus* alkaloids, and literature search only revealed (-)-*N*-oxide-pseudocyclobuxine-D, reported by Vachnadze et al. from *Buxus colchica* [24].

Compound **3** was obtained as a white gum, and its molecular formula, determined from its LC/MS molecular adduct ions at *m*/*z* 217.2010 [M + 2H]^2+^ and 433.3840 [M + H]^+^, was identical to that of **2** (C_27_H_48_N_2_O_2_; Appendix A). The ^1^H- and ^13^C-NMR spectroscopic data (Table 3 and Table 4, respectively; Appendix A) also showed great similarity between **3** and **1**. However, unlike the methylene group at C-16 in **1**, compound **3** exhibited signals corresponding to a hydroxy methine group at δ_H_ = 4.30 ppm (ddd, J = 9.6, 7.5, 2.2 Hz)/δ_C_ = 77.1 ppm, which showed an HMBC correlation with C-20. In addition, the methylene group at C-15 appeared downfield shifted at δ_C15_ = 47.9 ppm compared to 36.5 ppm in compound **1**, confirming the presence of an electron withdrawing group at C-16. The NOE correlations of H-16 with H-15β (δ_H_ = 2.03 ppm), H-20 (δ_H_ = 3.58 ppm) and, most importantly, CH_3_-18 (β, δ_H_ = 1.08 ppm) confirmed the relative configuration at this position (H-16β, OH-16α). Based on these data, compound **3** was, thus, unambiguously identified as 16α-hydroxycycloprotobuxoline-C.

The molecular formula of compound **5**, which was obtained as a colorless gum, was determined by the LC/MS as C_37_H_58_N_2_O_6_, from its quasimolecular ions at *m*/*z* 314.2334 [M + 2H]^2+^ and 627.4483 [M + H]^+^ (Appendix A). Sharing the same core nortriterpenoid framework as compound **1**, the ^1^H- and ^13^C-NMR spectroscopic data (Table 3 and Table 4, respectively; Appendix A) of compound **5** showed replacement of the C-29 methyl group by a methylene bearing an oxygen substituent (δ_H29_ = 4.62 ppm (d, J = 13.0 Hz); 4.17 ppm (d, J = 13.0 Hz)/δ_C29_ = 64.9 ppm), and additional aromatic and methoxy signals that were assigned to a trimethoxybenzoate moiety. The HMBC correlations from the C-29 methylene protons to the ester carbonyl confirmed attachment of the trimethoxybenzoate group at this position. It is important to note that ^13^C-NMR and X-Ray diffraction studies on *Buxus* alkaloids have shown that the C-4α methyl (C-29) undergoes preferential oxidation compared to the C-4β methyl (C-30), indicating the need to revise previous β-configuration of hydroxymethylene or other derivatives attached to C-4 unless evidence to the contrary is available [25,26]. Furthermore, the NOESY spectrum of compound **5** showed strong correlations of the C-30 methyl protons with both C-2 and C-19 protons. In light of these data, the stereochemical configuration at C-4 was assigned. To the best of our knowledge, benzoates and benzamides occurring frequently, trimethoxybenzoate esters have not previously been reported among the alkaloids of *Buxus* species. This structural modification expands the chemical diversity observed within *Buxus* alkaloids; accordingly, compound **5** was assigned the new trivial name 29-trimethoxybenzoyloxy cycloprotobuxoline-C.

Compound **6**, whose molecular formula was established from its LC/MS molecular adduct ions at *m*/*z* 202.1918 [M + 2H]^2+^ and 403.3722 [M + H]^+^ as C_26_H_46_N_2_O (Appendix A), was obtained as a yellow gum. The signals of the ^1^H- and ^13^C-NMR spectra (Table 3 and Table 4, respectively; Appendix A) were, for the most part, in close agreement with those of derivatives **1**–**3**, indicating the same nortriterpenoid core and substitution pattern in the side-chain at C-17. The key difference was the presence of only two *N*-methyl singlets (δ_H33/34_ = 2.90 s, 3H; 2.72 s, 3H) instead of three. An upfield shift in C-3 (δ_C3_ 67.2 ppm) indicated a primary amino group at this position, replacing the secondary (monomethylamino) group observed in the previous derivatives (δ_C3_ 76.7–76.8 ppm). The configuration of the amino substituent was unchanged in comparison to **1**–**3**, as deduced from NOESY correlations between H-3 and H-5, together with a large coupling constant (J = 10.5 Hz) between H-3 and the equatorial H-2. Based on these data, compound **6** was identified as *N*_3_-demethylcycloprotobuxoline-C.

For compound **7**, isolated as a white gum, the molecular formula was determined as C_26_H_46_N_2_O_2_ by its LC/MS, which showed molecular adduct ions at *m*/*z* 210.1910 [M + 2H]^2+^ and 419.3691 [M + H]^+^ (Appendix A). Most signals in the ^1^H- and ^13^C-NMR spectra (Table 3 and Table 4, respectively; Appendix A) were generally similar to those of **6**. The principal difference was observed at C-16 of **7**, where the methylene group present in compound **6** was replaced by a hydroxy methine. As already observed with compound **3**, it was shown that the OH group occupies the α-position at C-16. Accordingly, compound **7** was identified as 16α hydroxy-*N*_3_-demethylcycloprotobuxoline-C.

Compounds **8a** and **8b** were obtained as a mixture inseparable by preparative HPLC. The LC/MS analysis (Figure 2A–C) revealed two partly separated peaks with identical molecular masses (at *m*/*z* 216.1896 [M + 2H]^2+^/431.3678 [M + H]^+^ and *m*/*z* 216.1897 [M + 2H]^2+^/431.3671 [M + H]^+^, respectively) and the same molecular formula (C_27_H_46_N_2_O_2_). Compared to compound **1**, the ^1^H- and ^13^C-NMR spectroscopic data of the two compounds, (Table 3 and Table 4, respectively; Appendix A), indicated the presence of a monomethylamino group at C-20 and a formamide group at C-3. The NMR spectra of **8a** and **8b** were largely superimposable, differing primarily in the chemical shifts in the C-3 methine and the amide-linked *N*-methyl group, C-31. Furthermore, MS/MS fragmentation patterns confirmed identical core structures in both compounds (Figure 2B,C), with characteristic neutral losses of the C-3 methylformamide, C-2 hydroxyl, and C-20 methylamino substituents. Analysis of the ^1^H–^1^H coupling constants between H-2 and H-3 showed identical J values in both **8a** and **8b**, ruling out epimerization or configurational isomerism at C-3 (Figure 2D).

Instead, the observation of distinct ^1^H and ^13^C resonances for the *N*_3_-methyl and C-3 methine groups strongly supported the presence of rotational isomers (rotamers) arising from restricted rotation about the C–N amide bond at C-3. In the ^13^C-NMR of **8a** (*trans* rotamer) the amide methyl was more shielded (δ_C31_ = 29.4 ppm) and the C-3 more deshielded (δ_C3_ = 74.2 ppm) compared to **8b** (*cis* rotamer; δ_C31_ = 33.7 ppm; δ_C3_ = 66.6 ppm) consistent with reported shielding/deshielding effects in *N*-methylated amides [27]. The presence of the axial hydroxyl group at C-2 may influence the rotamer equilibrium and stabilization through intramolecular H-bonding or steric interactions, enhancing the NMR distinction between the two forms [28]. Based on the integrals in the ^1^H-NMR spectrum, the proportion of the two compounds was calculated to be 80% (*trans* rotamer) and 20% (*cis* rotamer). To the best of our knowledge, this represents the first report of a C-3 *N*-methyl formamide substitution and spectroscopic evidence for hindered amide rotation in a *Buxus* alkaloid. Based on the above-mentioned data, and the relation to compound **4**, compounds **8a** and **8b** were assigned as cylcoprotobuxoline-D *N*_3_-*trans*- and cylcoprotobuxoline-D *N*_3_-*cis*-formamide, respectively.

Compounds **9a** and **9b** were also obtained as a mixture. LC/MS analysis (Appendix A) showed identical molecular masses (at *m*/*z* 231.1955 [M + 2H]^2+^/461.3808 [M + H]^+^ and *m*/*z* 231.1940 [M + 2H]^2+^/461.3775 [M + H]^+^, respectively) and the same molecular formula (C_28_H_48_N_2_O_3_). Most signals in the ^1^H- and ^13^C-NMR spectra (Table 3 and Table 4, respectively; Appendix A) closely resembled those of **8a/8b**. In contrast to a monomethylated amine in **8a**/**8b**, **9a**/**9b** displayed signals of two *N*-methyl groups of a dimethylamino group at C-20 (δ_H_ = 2.81/2.96 ppm, (s, 6H)/δ_C_ = 36.6/43.7 ppm) and a hydroxylated methine in position 16 (δ_H_ = 4.30 ppm (ddd, J = 9.5, 7.6, 2.2 Hz)/δ_C_ = 77.2 ppm), both showing HMBC correlations with C-20 and adjacent carbons. Thus, in contrast to compounds **8a**/**8b**, the structures of **9a**/**9b** possess a tertiary amine (dimethylamino group) at C-20 instead of a secondary amine (monomethylamino group) and a hydroxyl group at C-16 instead of a methylene. As with **8a** and **8b**, the two analogues differed mainly in the chemical shifts in the C-3 methine and the amide-linked *N*-methyl carbon, supporting their assignment as rotational isomers about the C–N amide bond at C-3. Thus, compounds **9a** and **9b** were identified as the formamide derivatives of compound **3** and hence consistently named 16α-hydroxycycloprotobuxoline-C *N*_3_-*trans*-formamide and 16α-hydroxycycloprotobuxoline-C *N*_3_-*cis*-formamide.

The molecular formula of compound **11**, obtained as a white gum, was established from its LC/MS quasimolecular ions at *m*/*z* 216.1937 [M + 2H]^2+^ and 431.3721 [M + H]^+^, as C_27_H_46_N_2_O_2_ (Appendix A). The complete ^1^H- and ^13^C-NMR data of **11** (Table 5 and Table 6, respectively; Appendix A) was generally similar to those of **3**. The key difference was the presence of two conspicuous downfield resonances (δ_H_ = 5.71 ppm (d, J = 10.7 Hz)/δ_C_ = 126.4 ppm) and (δ_H_ = 5.57 ppm (ddd, J = 10.7, 6.2, 3.2 Hz)/δ_C_ = 130.9 ppm) attributable to a double bond, specifically Δ^6,7^ in a *Buxus* alkaloid [12,18]. Structurally, compound **11** was identified as the 6,7-dehydro derivative of **3** but also an analogue of the known compound **10** (cyclonataminol), from which it differs in that it bears a monomethylamino moiety at C-3 instead of a dimethylamino group. Consequently, compound **11** was named *N*_3_-demethyl cyclonataminol.

Compound **12** was obtained as a yellow gum, and its molecular formula (C_27_H_46_N_2_) was determined by LC/MS analysis which displayed quasimolecular ions at *m*/*z* 200.1954 [M + 2H]^2+^ and 399.3785 [M + H]^+^ (Appendix A). In contrast to the previous compounds all bearing a hydroxy group at C-2, the ^1^H- and ^13^C-NMR data of **12** (Table 5 and Table 6, respectively; Appendix A) displayed signals of a methylene group at this position (δ_H2_ = 1.91 ppm, m; 2.11 ppm, m/δ_C2_ = 21.6 ppm). This assignment was confirmed by the ^1^H/^1^H COSY correlations between H-2 and both H-1 and H-3 as well as all other spectral features. The structure of compound **12** was thus established as a derivative of cyclovirobuxeine-B (compound **14**) [12,29], differing by the presence of a methylene group at C-16 instead an OH substituent. Based on these findings, compound **12** was named deoxycyclovirobuxeine-B.

The molecular formula of compound **15**, obtained as a colorless gum, was determined as C_27_H_46_N_2_O by LC/MS analysis which showed quasimolecular ions at *m*/*z* 208.1943 [M + 2H]^2+^ and 415.3736 [M + H]^+^ (Appendix A). The structural similarity of this alkaloid to the above-described compounds was indicated by its ^1^H- and ^13^C-NMR data (Table 5 and Table 6, respectively; Appendix A), which showed the typical resonances for the fully saturated nortriterpenoid (norcycloartane) core, including a tertiary C-methyl, a cyclopropyl methylene, three quaternary C-methyls and no olefinic protons, along with a methylamino moiety attached to C-20. However, of interest was the observation of a set of distinctive AB doublets centred at δ_H_ = 3.56 ppm, (d, J = 11.1 Hz) and 3.90 ppm, (d, J = 11.1 Hz) assignable to C-29 methylene protons, while another pair of AB doublets at δ_H_ = 4.40 ppm (d, J = 8.6 Hz) and 5.03 ppm (d, J = 8.6 Hz) corresponded to methylene protons (H-31) adjacent to the C-3 nitrogen [30,31]. HMBC correlations from H-3 to both methylene pairs confirmed the presence of a tetrahydro-1,3-oxazine ring in ring A. Comparison with literature data indicated a close relationship between compound **15** to deoxycyclobuxoxazine A [32], from which it differs only by the replacement of the C-20 dimethylamino group with a methylamino group. Compound **15** was therefore named *N*_20_-demethyl deoxycyclobuxoxazine A.

The LC/MS analysis of **16** (Appendix A) displayed a singly charged quasimolecular ion at *m*/*z* 374.3115 [M + H]^+^ indicating the presence of a monobasic alkaloid. The molecular formula of **16** was thereby established as C_24_H_39_NO_2_. Its ^1^H- and ^13^C-NMR spectra (Table 5 and Table 6, respectively; Appendix A) revealed signals characteristic of the C-19 cyclopropane methylene, a C-3 monomethylamino substituent, and two quaternary C-methyl groups at C-18 and C-28. Additional resonances included a sharp singlet germinal to a carbonyl carbon at δ_H_ = 2.16 ppm, (s, 3H)/δ_C_ = 31.5 ppm, assigned to C-21, and a methoxy singlet (δ_H_ = 3.18 ppm, (s, 3H)/δ_C_ = 57.5 ppm) attributed to C-16, whose position was confirmed by HMBC correlations between the methoxy protons and the C-16 methine carbon. In contrast to the 9,19-cyclo-5α-pregnane derivatives possessing a quaternary C-4 (**1–15**), compound **16** exhibited a methylene group at C-4 (δ_H_ = 1.97 ppm, m; 1.18 ppm, (d, J = 11.8 Hz)/δ_C_ = 35.5 ppm), which was confirmed by its vicinal correlations with both H-3 and H-5 in the COSY spectrum, as well as its HMBC correlations with C-2, C-3, C-5 and C-10 (Appendix A). To the best of our knowledge, such a compound featuring an unsubstituted C-4 methylene within an otherwise norcycloartanoid carbon skeleton has not been previously described among the *Buxus* alkaloids. The vast majority of *Buxus* alkaloids reported to date feature either a 4,4-dimethyl substitution (often hydroxylated or esterified) or sometimes an exomethylene moiety (C = CH_2_) at this position. It is worth noting, that the presence of this compound in the crude extract confirmed it as a true natural plant metabolite. All spectral data led to the unambiguous assignment of the structure **16**. For this unusual new aminosteroid alkaloid with a previously unknown carbon scaffold, we chose the generic name obtusibuxeine A.

The LC/MS spectrum (Appendix A) of compound **17** showed quasimolecular ions at *m*/*z* 297.1921 [M + 2H]^2+^ and 593.3679 [M + H]^+^ corresponding to the molecular formula, C_35_H_48_N_2_O_6_ and indicating the presence of 13 double bond equivalents in the molecule. The intensity of the singly charged adduct was clearly higher than that of the doubly charged adduct, suggesting two nitrogen atoms with significantly different basicities. The observed fragmentation pattern revealed some key structural features of compound **17**. The fragment ion at *m*/*z* = 575 [M − (H_2_O)]^+^ arose from the loss of a hydroxyl group as a water molecule, while the subsequent fragment at 533 *m*/*z* [M − (CH_3_COOH)]^+^ was attributed to the loss of an acetate group as an acetic acid unit. Furthermore, the fragment signals at *m*/*z* = 72 and 105 indicated presence of an *N*,*N*-dimethylamino side chain at C-17 [C_4_H_10_N]^+^ and a benzoyl ion [C_7_H_5_O]^+^ corresponding to a benzamide moiety at C-3, respectively [33,34]. Based on the NMR spectroscopic analysis (Table 5 and Table 6; Appendix A) the ^13^C-NMR spectrum exhibited resonances characteristic to a 9(10→19) *abeo*-5α-pregnane framework [12,35,36], including signals for the allylic C-19 methylene carbon (δ_C19_ = 45.2 ppm), and olefinic carbons at C-1 (δ_C1_ = 133.8 ppm), C-2 (δ_C2_ = 130.5 ppm) and C-11 (δ_C11_ = 126.1 ppm). In the ^1^H-NMR spectrum, the C-29 methylene protons appeared as a set of AB doublets at δ_H29_ = 3.70 ppm and 3.62 ppm, (d, J = 8.6 Hz), while C-10 appeared as a quaternary carbon at δ_C10_ = 81.1 ppm. A combination of COSY and HMBC correlations indicated an ether linkage between C-29 and C-10 in accordance with previous findings observed in the related compound *O*^10^-natafuranamine [18]. Additional aromatic signals were assigned to a benzamide moiety, which was located at C-3 based on the HMBC cross signal of the H-3 (δ_H3_ = 4.71 ppm) with the carbonyl carbon (δ_C1″_ = 170.3). Two downfield methine protons geminal to an acetoxy group (δ_H_ = 5.05 ppm) and an OH group (δ_H_ = 4.02 ppm) were due to H-6 and H-7, respectively. Furthermore, the position of the acetoxy group at C-6 was confirmed by the cross peaks in the COSY (H-6 to H-5 and H-7) and HMBC (H-6 to C-1′) spectra, respectively. A multiplet signal at δ_H_ = 4.33 ppm was consistent with a C-16 methine proton, geminal to an OH group. The assignments at various stereo centres were based on the NOESY correlations, together with chemical shifts, coupling constants and biogenetic comparisons with literature data [18]. The a-orientation of the oxygenated C-4 methyl (C-29 methylene) was based on the NMR and X-ray diffraction studies by Sangre et al. and Guilhem, respectively [25,26]. Consistent with the 9,19-cyclo-5α-pregnanes, the C-16 hydroxyl group was assigned an α-orientation based on NOE correlations of H-16 with both H_3_-18 and H-20. The methine proton at C-6 exhibited as a double doublet (dd) showing one large coupling constant (11.3 Hz) with the axial H-5 and another small coupling constant (1.7 Hz) with H-7, indicating different spatial orientations for H-5 relative to H-6 and a similar orientation for H-6 and H-7. The measured dihedral angles of −178.0^0^ between H-5 and H-6, and 77.8^0^ between H-6 and H-7 were consistent with the observed coupling constants (see Figure 3, right 3D model, B). Additionally, NOE correlations were observed between H-7 and H_2_-15, which corroborated only with an a-oriented C-7 OH group, ruling out β-orientation (Figure 3, right vs. left 3D model, B vs. A). Furthermore, the observed NOE between H-7 and the a-oriented H_3_-28 was confirmed to be possible with both orientations of H-7 (Figure 3, right vs. left 3D model, B vs. A), that is, not to contradict the postulated configuration. Taken together, the spectral data led us to unambiguously elucidate structure **17** shown in Figure 1 and structure **17-B** in Figure 3 for this new compound, which was generically named *O*^10^-obtusifuranamine-A.

The molecular formula of compound **18** was established from the LC/MS spectrum (Appendix A) as C_40_H_50_N_2_O_6_ on the basis of the quasimolecular ions (*m*/*z* 328.1977 [M + 2H]^2+^, 655.3817 [M + H]^+^) and thus included 17 double bond equivalents. The molecular mass of **18** differed from that of **17** by 62 Da and exhibited identical MS/MS fragmentation, suggesting replacement of the acetate group by a benzoate moiety. Consistent with the LC/MS data, evaluation of NMR spectra (Table 5 and Table 6; Appendix A) indicated that most signals were identical to those of compound **17**. The key difference was additional aromatic signals, which were assigned to a benzoate group bound to the oxygen at C-6. The stereochemical assignments for various stereo centers were established in a similar manner to that used for **17.** Therefore compound **18** was identified as the benzoate analog of **17** and named *O*^10^-obtusifuranamine-B.

Compound **19** had a molecular formula of C_40_H_50_N_2_O_5_ as indicated by the LC/MS quasimolecular ions at *m*/*z* 320.1993 [M + 2H]^2+^ and 639.3802 [M + H]^+^ (Appendix A), representing a difference of -16 Da from compound **18**. Comprehensive comparison of its NMR data (Table 5 and Table 6; Appendix A) with those of compound **18** revealed a high degree of similarity in the observed signals. The key deviation occurred in the chemical shifts at position C-16, where the hydroxymethine group of **18** was replaced by a methylene group. This conclusion was supported by the COSY correlations of the C-16 methylene protons with adjacent protons, as well as the HMBC correlation from H-20 (δ_H20_ = 3.44 ppm) to C-16 (δ_C16_ = 26.0 ppm). Compound **19** was thus elucidated as 16-deoxy-*O*^10^-obtusifuranamine-B.

Compound **21** afforded the molecular formula of C_35_H_48_N_2_O_5_ as indicated by its LC/MS quasimolecular ions at *m*/*z* 289.1940 [M + 2H]^2+^ and 577.3728 [M + H]^+^, which differed from compound **17** by 16 Da (Appendix A). Based on its NMR spectroscopic data (Table 5 and Table 6; Appendix A), compound **21** was confidently assigned to the *Buxus* alkaloids class characterized by a 9(10→19) *abeo*-5α-pregnane backbone. In analogy to **17**, compound **21** showed characteristic NMR resonances corresponding to a C-3 benzamide moiety, a C-6 acetoxy substituent, a quaternary C-10, a hydroxylated C-16, an allylic C-19 methylene group, and a dimethylamino side chain. However, in contrast to compounds **17**–**19**, compound **21** lacked an ether linkage between C-29 and C-10. Instead, it exhibited distinct NMR signals corresponding to two geminal methyl groups at C-4 (δ_H29_ = 1.09 ppm, (s, 3H)/δ_C29_ = 28.2 ppm; δ_H30_ = 1.08 ppm, (s, 3H)/δ_C30_ = 18.9 ppm). The C-6 methine proton (δ_H6_ = 4.99 ppm), geminal to the C-6 acetoxy group, exhibited COSY cross peaks to H-5 (δ_H5_ = 2.67 ppm) and an oxygenated methine assigned to C-7 (δ_H7_ = 3.89 ppm). The latter showed vicinal coupling with H-8 (δ_H8_ = 2.26 ppm). Analysis of the HMBC spectrum confirmed the presence of an epoxide moiety between C-7 and C-10, as evidenced by a distinct three bond (^3^*J*_CH_) correlation from H-7 to C-10. The epoxidated *abeo*-pregnane type *Buxus* alkaloids reported to date have featured epoxide linkages between C-29/C-10, as observed in compounds **17**–**19**, C-29/C-2 and C-1/C-10 as in compound **20,** or at C-29/C-6 or C-9/C-11 positions [18,34,37]. Compound **21**, therefore, represents the first example of this structural type bearing an epoxide bridge between C-7 and C-10. As already demonstrated for compound **17**, the substituents at C-6, C-7 and C-16 were confirmed to adopt the α-orientation. To further validate the α-configuration of the epoxide, 3D molecular models were generated and various interproton distances measured to support the proposed stereochemistry. As depicted in Figure 4 (right vs. left 3D model, B vs. A), the epoxide bridge between C-7 and C-10 must adopt the α-configuration, as only this spatial arrangement positions H-6 sufficiently close to H-8 and H-19b, to account for the observed NOE correlations. Additionally, the NOE correlations of H-7 with both H-15a and H_3_-28 were similarly substantiated, further supporting the proposed stereochemistry. Based on these data, compound **21** was assigned the structure depicted in Figure 1 and structure **21-B** in Figure 4 and was named obtusiepoxamine A.

The molecular formula of compound **22** was determined as C_35_H_50_N_2_O_5_ on the basis of the LC/MS quasimolecular ions at *m*/*z* 290.2016 [M + 2H]^2+^ and 579.3887 [M + H]^+^ (Appendix A). The NMR spectroscopic data of compound **22** (Table 5 and Table 6; Appendix A), suggested the presence of another *Buxus* alkaloid with a 9(10→19) *abeo*-5α-pregnane skeleton. This structure closely resembles those of compounds **17**–**19** with the key distinction being the presence of a double bond between C-1 and C-10, rather than between C-1 and C-2 as observed in the latter. This structural assignment was supported by two multiplets corresponding to geminal methylene protons at δ_H_ = 2.18 ppm and 2.34 ppm (δ_C_ = 30.0 ppm in the ^1^H/^13^C-HSQC spectrum), which displayed vicinal correlations with the methine protons at C-1 and C-3 in the COSY spectrum, thereby confirming their position at C-2. Furthermore, the quaternary carbon at C-10 resonated significantly downfield (δ_C_ = 136.8 ppm), consistent with the presence of a C-1/C-10 double bond [33,38]. As observed in **21**, compound **22** also exhibited signals corresponding to two geminal methyl groups at C-4 (δ_H29_ = 1.08 ppm, (s, 3H)/δ_C29_ = 28.9 ppm; δ_H30_ = 1.00 ppm, (s, 3H)/δ_C30_ = 22.1 ppm). The configuration of the stereocentres at C-6, C-7 and C-16 was confirmed by 3D molecular modelling, which showed full agreement with all NMR spectroscopic data. Based on these spectroscopic analyses, the structure of this new alkaloid was unambiguously established and designated as obtusidienolamine-A.

For compound **23**, the molecular formula was determined to be C_35_H_50_N_2_O_4_, based on its LC/MS quasimolecular ions at *m*/*z* 282.1999 [M + 2H]^2+^ and 563.3801 [M + H]^+^ (Appendix A). The majority of signals for compound **23** in the NMR spectra (Table 5 and Table 6; Appendix A) were similar to those observed for compound **22**. The only structural difference was observed at C-16, where the hydroxy methine group present in compound **22** was replaced by a methylene group. Consequently, compound **23** was identified as 16-deoxyobtusidienolamine-A.

The LC/MS spectrum of compound **24** (Appendix A) displayed a singly charged molecular adduct ion at *m*/*z* 368.2660 [M + H]^+^, corresponding to the molecular formula C_24_H_33_NO_2_ and indicating the presence of nine double bond equivalents.

The NMR spectroscopic data (Table 7; Appendix A) revealed the typical structural characteristics associated with a nortriterpenoid *Buxus* alkaloid. The ^1^H-NMR spectrum showed multiple methylene and methine resonances, along with signals corresponding to five methyl groups (δ_H18_ = 1.14 ppm, δ_H21_ = 1.85 ppm, δ_H28_ = 0.85 ppm, δ_H29_ = 1.34 ppm, and δH_30_ = 1.05 ppm), while no signals attributable to methylamino groups were observed. An olefinic methine proton resonance at δ_H_ = 6.51 ppm (q, J = 6.1 Hz), correlated with a carbon signal at δ_C_ = 131.0 ppm in the ^1^H/^13^C-HSQC spectrum suggesting a geminal position to a methyl group, and was assigned to C-20. The ^13^C-NMR in combination with APT and DEPT-135 spectra, revealed 24 carbon signals, comprising five methyl groups, six methylene groups, five methine carbons (including one olefinic at δ_C_ = 131.0 ppm). Additionally, eight quaternary carbons were observed among which were, one olefinic at δ_C_ = 148.5 ppm and two carbonyl carbons at δ_C_ = 206.6 and 211.3 ppm. The two carbonyl groups and one double bond accounted for three degrees of unsaturation, thereby implying the existence of six rings within the molecule.

The positions of the two carbonyl groups at C-6 (δ_C_ = 206.6 ppm) and C-16 (δ_C_ = 211.3 ppm) were unequivocally established by HMBC correlations from H-5, H-7, and H-8 to C-6 and from H-15 and H-20 to C-16, respectively (Figure 5). Further analysis of the NMR data deduced the presence of an unprecedented C-3/C-10 amino bridge, consistent with the absence of a methylamino substituent in the molecule. This conclusion was supported by HMBC correlations (^3^*J*_CH_) from H-3 (δ_H3_ = 3.81 ppm) to C-1 (δ_C_ = 30.0 ppm), C-5 (δ_C_ = 62.1 ppm), C-10 (δ_C_ = 81.4 ppm), and C-30 (δ_C_ = 23.8 ppm). Additionally, COSY correlations between H-3 and H-2, and between H-2 and H-1 provided further evidence for this connectivity (Figure 5). The cyclopropane methylene protons exhibited downfield chemical shift values and elevated coupling constants (δ_H19a_ = 1.18 ppm, (dd, J = 10.9, 6.3 Hz) and δ_H19b_ = 1.30 ppm, (d, J = 6.6 Hz)) compared to those observed in the 9,19-cyclo-5α-pregnane (**1–16**) derivatives, indicating a different mode of ring closure and thus a unique position of the cyclopropane moiety. Additionally, COSY cross-peaks between H-11 (δ_H11_ = 1.58 ppm/δ_C_ = 22.6 ppm) and both C-19 and C-12 methylene groups, and HMBC correlations (^3^*J*) from H_2_-19 to C-8 (δ_C_ = 40.4 ppm), C-10 (δ_C_ = 81.4 ppm), and C-12 (δ_C_ = 30.4 ppm) further confirmed the position of the cyclopropane ring within the molecule. Furthermore, H_2_-1, H-8, H_2_-19 and H_2_-12, all showed ^2^*J* or ^3^*J* correlations with the quaternary C-9 (δ_C_ = 22.6 ppm). These diagnostic correlations unequivocally confirmed that the cyclopropane ring is closed between C-9, C-19 and C-11. Until now, all previously known *Buxus* alkaloids bearing a cyclopropane moiety possess the norcycloartanoid carbon skeleton, wherein the cyclopropane ring typically comprises C-9, C-10 and C-19. An exception of this pattern is spirofornabuxine, which features a unique rearranged spirocyclic scaffold [39].

The relative stereochemistry of **24** was established through analysis of its NOESY spectrum (Appendix A), complemented by and biosynthetic considerations of *Buxus* steroidal alkaloids. The NOE correlations of H_2_-19 with H-8 and H_3_-18 suggested a β-orientation of that the cyclopropane ring. Furthermore, the correlations between H-5, and methyl groups H_3_-28 and H_3_-29 suggested a “*syn*” spatial relationship, which, based on biosynthetic grounds, is consistent with an α-orientation. Nearly all previously reported *Buxus* alkaloids, as well as all other compounds characterized in this study possess a nitrogen substituent at C-3 in the *β*-configuration. Only a limited number of *Buxus* alkaloids featuring a 3*α*-amino substituent have been documented [40,41]. In compound **24**, the amino-substituent was assigned the more common β-orientation on the basis of the NOE correlations between the a-oriented H_3_-29 with H-2a. This structural arrangement is only feasible if the amino bridge between positions C-3 and C-10 adopts a β-orientation rather than an α-orientation (Figure 5, right vs. left 3D model, B vs. A). Additionally, a distinct NOE correlation between H_3_-30 (β) and H-7b was observed, which would not be possible in the stereoisomer bearing an α-oriented amino group. Compound **24** was thus unambiguously assigned the structure depicted in Figure 1 and structure **24-B** in Figure 5. 

Notably, this structurally unique alkaloid was detected in the LC/MS chromatogram of the crude extract, confirming its presence as a native component of *Buxus obtusifolia* with a novel structural framework. It is likely that compound **24**, like all other *Buxus* alkaloids, is biosynthetically derived from cycloartenol, the common precursor of all *Buxus* alkaloids [42,43]. However, its formation may involve a previously uncharacterized pathway, potentially involving a rearrangement through an *abeo*-pregnane type intermediate, as observed in compounds **17–23**. This novel *Buxus* alkaloid was named obtusiaminocyclin.

### 2.3. Antiprotozoal Activity of the Aminosteroids Isolated from Buxus Obtusifolia

The isolated compounds were assessed for their in vitro antiplasmodial and antitrypanosomal properties against *Trypanosoma brucei rhodesiense* (*Tbr*), and *Plasmodium falciparum* (*Pf*), respectively. In order to determine their selectivity toward the target parasites, their cytotoxicity against L6 rat skeletal myoblasts was also evaluated (Table 8). As the alkaloids were isolated as mono- or *bis*-trifluoroacetate salts (depending on the number of basic amino groups), the molecular masses of the salts were used to calculate the molar IC_50_ values. It is important to note that sodium trifluoroacetate was separately tested and found to be inactive against the target parasites in a recent study by our research group [14]. Among the 9,19-cyclo-5α-pregnanes (**1–16**), the new compounds **6** and **12** demonstrated the highest antitrypanosomal activity with IC_50_ values of 0.9 µmol/L and 0.8 µmol/L and SI values of 53 and 108, respectively. Albeit with moderate potency, the *N*-oxide derivative (compound **2**), demonstrated 2-fold the antitrypanosomal activity (IC_50_ = 3.8 µmol/L vs. IC_50_ = 6.7 µmol/L) and 3-fold the antiplasmodial activity (IC_50_ = 5.2 µmol/L vs. IC_50_ = 28 µmol/L) as compared to the unoxidized parent compound **1**.

The 9(10→19) *abeo*-5α-pregnane derivatives (**17–23**) generally exhibited low antitrypanosomal activities with the exception of compounds **19** (IC_50_ = 2.0 µmol/L) and **21** (IC_50_ = 2.2 µmol/L), which showed moderate activities against *Tbr*. It is worth noting that cyclovirobuxeine-B (**14**), previously reported by our research group to possess significant antitrypanosomal activity [12,21], demonstrated considerably reduced antiprotozoal activity in this study compared to its newly characterized derivative, deoxycyclovirobuxeine-B (**12**).

Remarkable antiplasmodial activities were recorded for compounds **4** (IC_50_ = 1.1 µmol/mL) and, especially **5** (IC_50_ = 0.5 µmol/L) with the trimethoxy benzoate moiety. Additionally, compound **6** (IC_50_ = 3.0 µmol/L) exhibited moderate antiplasmodial activity, and the highest selectivity index against *Pf* (SI = 17). The rest of the alkaloids showed low antiplasmodial activity with IC_50_ values ranging from 5.4 to 49 µmol/L.

The in vitro antiprotozoal activity data enabled a preliminary structure–activity relationship (SAR) analysis of the isolated compounds. Overall, the 9,19-cyclo-5α-pregnanes (**1–16**) exhibited stronger antiprotozoal activity and more favourable selectivity indices compared to the 9(10 → 19) *abeo*-5α-pregnane analogues (**17–23**). Furthermore, within the 9,19-cyclo-5α-pregnane series, it was observed that di- or monomethylation at both the C-3 amino group and C-20 significantly influence the antiprotozoal potency. These observations are consistent with previous findings reported by our group in the study of *B. sempervirens* alkaloids [12]. Notably, the most active compound identified in the present study (compound **5**) features a bulky trimethoxy benzoate substituent at C-29 position, while the most potent compounds against both *Pf* and *Tbr* in the study on *B. sempervirens*, namely *O*-benzoyl-cycloprotobuxoline-D and C-16 cyclomicrophyllidine-B contained unsubstituted benzoate groups at either C-2 or C-16, respectively. Interestingly, a recent study by Schäfer et al. on *Pachysandra terminalis*, a closely related species within the Buxaceae family afforded an aminosteroid alkaloid bearing a benzoate group at C-4 (3α,4α-Diapachsanaximine A). This compound showed very strong activities against both protozoan parasites [14]. Given the proposed similarity in the mechanisms of action against these two distinct parasites [14], it is evident that the presence of a benzoate moiety on either ring A or D (whether substituted or unsubstituted) in combination with *N*-methylation at C-3 and C-20 appears to be a critical structural feature contributing to antiprotozoal activity.

## 3. Materials and Methods

### 3.1. Plant Material

The leaves of *B. obtusifolia* (Mildbr.) Hutch. were collected from the Gongoni forest Kenya (04°24′38.2″ S 039°28′34.6″ E) in May 2022. The plant material was identified by Mr. Patrick Mutiso, a taxonomist at the Faculty of Science and Technology, University of Nairobi. The voucher specimens were deposited at both the University of Nairobi Herbarium (UoN_JM 2022_002) and at the Institute of Pharmaceutical Biology and Phytochemistry, University of Münster (IPBP 916-TS_JM_2022_002). The plant material was air-dried under shade at room temperature to constant weight and then ground into fine powder using a mill.

### 3.2. Extraction of Buxus Obtusifolia Leaves

In a Soxhlet apparatus, the powdered plant material (680 g) was exhaustively extracted in three equal parts with 1.5 L dichloromethane (CH_2_Cl_2_) for each part for 36 h until the supernatant was clear. The extracts were combined and evaporated in vacuo at 40 °C to obtain 35.3 g of crude extract, translating to 5.2% yield. In order to separate the alkaloids from the crude extract, an acid-base extraction was performed. For each batch of extraction, 5 g of the extract was once again dissolved in 100 mL of CH_2_Cl_2_. Using a separatory funnel, this was extracted seven times using 30 mL of diluted hydrochloric acid (aq., 1 M). The organic CH_2_Cl_2_ phases were combined and evaporated using a rotary evaporator at 40 °C to yield a total of 10.2 g of the neutral fraction. The aqueous phases were alkalized to ≈ pH 10 with sodium hydroxide solution (aq., 2 M) and exhaustively extracted with 400 mL (6 times) CH_2_Cl_2_ to yield, after evaporation, a total of 15.4 g of the alkaloid fraction (2.3% yield) which was then kept in a refrigerator at 4 °C for subsequent work.

### 3.3. Isolation of Alkaloids from Buxus Obtusifolia Leaf Extract

Using a CPC-250 (Gilson, Limburg, Germany) chromatography system, a portion of (5 g of the alkaloid fraction was separated using the centrifugal partition chromatography (CPC) technique that was previously utilized for *B. sempervirens* in our working group [11], with minimal modifications. This was accomplished using a biphasic solvent system with iso-hexane/ethyl acetate (8/2; *v*/*v*) as the upper phase and CH_3_OH/H_2_O/propan-2-ol (8/2/1; *v*/*v*/*v*) as the lower phase. The eluent system was equilibrated in a separatory funnel overnight and sonicated before the experiment. The alkaloid fraction (5 g in 8 portions of 0.5–1 g) was dissolved in 8 mL (6 mL upper phase + 2 mL lower phase) and filtered using a 0.45-micron syringe filter. In ascending mode (1200 rpm, 2 mL/min), portions of 4 mL were collected into test tubes. After termination of the elution mode (624 mL) the lower phase was also fractionated and collected into test tubes by stopping the rotation and increasing the flow rate (6 mL/min) at the same time. The collected fractions were monitored on pre-coated silica gel 60 F_254_ thin layer chromatography (TLC) plates, (Merck KGaA, Darmstadt, Germany) using a mobile phase of butan-1-ol:H_2_O:CH_3_COOH (10:3:1.5) (*v*/*v*/*v*) followed by spraying with Dragendorff’s reagent (bismuth subnitrate (0.85 g):H_2_O (40 mL): CH_3_COOH (10 mL):potassium iodide solution (40%; 20 mL)). Based on the TLC and LC/MS profiles, the eluates were combined in 16 subfractions (1–16). Of these, 12 were obtained from the upper phase while the lower phase afforded 4 subfractions (compare Figure 1).

CPC subfractions 3 (351.0 mg), 4 + 5 (79.6 mg), 6–8 (138.1 mg), 9–10 (195.3 mg), 11 (70.4 mg) and 15 (447.5 mg) were separated by prep-HPLC on a RP-18 phase (VP 250/21 Nucleodur C-18 HTec with a VP 10/16 Nucleodur C-18 HTec pre-column, Macherey-Nagel, Düren, Germany) using binary gradients of H_2_O (+0.1% TFA; A) and ACN (+0.1% TFA; B). The following gradient conditions were used in all separations: 5–15% of B (0.1–15 min), 15–25% of B (15–30 min), 25–40% of B (30–45 min), 40–50% of B (45–50 min), 50–100% of B (50–55 min), and 100% of B (55– 60 min) at a flow rate of 10 mL/min and a column temperature of 40 °C. The separation of CPC subfraction 3 yielded compound **1** (17.3 mg, t_R_ 21.5 min), **2** (1.3 mg, t_R_ 26.8 min), **6** (1.8 mg, t_R_ 19.9 min) and **13** (4.9 mg, t_R_ 24.6 min). prep-HPLC of CPC subfractions 4 and 5 resulted in isolation of compound **12** (5.3 mg, t_R_ 29.5 min) while compounds **7** (1.9 mg, t_R_ 16.8), **3** (2.1 mg, t_R_ 18.7 min), and **9a** + **9b** (1.2 mg, t_R_ 32.5) were obtained from CPC subfractions 6-8. Compounds **4** (24.0 mg, t_R_ 24.2), **10** (18.3 mg, t_R_ 26.7), **11** (1.2 mg, t_R_ 20.4), and **14** (3.1 mg, t_R_ 27.3), were distributed in CPC subfractions 9 and 10 in varying concentrations while compound **8a** + **8b** (1.4 mg, t_R_ 44.6) were isolated from CPC subfraction 11. The separation of CPC subfraction 15 resulted in isolation of compounds **5** (2.1 mg, t_R_ 34.2), **15** (4.6 mg, t_R_ 23.1), **16** (2.6 mg, t_R_ 48.5), **17** (2.2 mg, t_R_ 38.6), **18** (9.0 mg, t_R_ 54.3), **19** (7.2 mg, t_R_ 56.6), **20** (3.5 mg, t_R_ 47.2), **21** (5.9 mg, t_R_ 51.8), **22** (2.0 mg, t_R_ 53.7), and **23** (1.5 mg, t_R_ 55.1) **24** (2.8 mg, t_R_ 32.3).

### 3.4. Spectrometric and Spectroscopic Analysis

The crude extract, alkaloid fraction, CPC subfractions and the isolated compounds were analysed using a Bruker Daltonics micrOTOFQII time-of-flight mass spectrometer (Bruker Daltonics GmbH, Bremen, Germany) fitted with an Apollo electrospray ionization source in positive mode at 3 Hz over a mass range of *m*/*z* 50–1500 using the following instrument settings: nebulizer gas (N_2_), 5 bar; dry gas (N_2_), 9 L/min, drying temperature, 200 °C; capillary voltage, 4500 V; end plate offset -500 V; transfer time 100 µs; collision gas (N_2_); collision energy 20 eV. MS^2^ spectra were acquired using a collision energy of 40 eV and an isolation width of 5 *m*/*z* units. Internal calibration (Quadratic + High Precision Calibration (HPC) mode) was performed for each analysis using the mass spectrum of a 10 mM solution of sodium formate solution prepared in propan-2-ol/water/formic acid/1M NaOH (50:50:0.2:1, *v*/*v*/*v*/*v*), and introduced into the source during LC re-equilibration via a divert valve equipped with a 20 µL sample loop. The compounds were separated on a C-18 column (Dionex Acclaim RSLC 120, Thermo Fisher Scientific, Waltham, MA, USA) and detected using a Dionex Ultimate DAD-3000 RS (Thermo Fisher Scientific, Waltham, MA, USA) at a wavelength of 200–400 nm. Using a binary gradient of H_2_O (+0.1% formic acid; A) and Acetonitrile (+0.1% formic acid; B) at a flow rate of 0.4 mL/min and column temperature of 40 °C, the following method previously developed for *Buxus* alkaloids [11] was used: 0–1.88 min: linear from 15% B to 30% B; 1.88–7.88 min: linear from 30% B to 33% B; 7.88–9.9 min: linear from 33% B to 50% B; 9.9–9.93 min: linear from 50% B to 100% B; 9.93–15.88: isocratic 100% B; 15.88–15.98 min: linear from 100% B to 15% B; 15.98–20.0 min: isocratic 15% B. The following sample concentrations were used: 10 mg/mL (crude extract), 1 mg/mL (alkaloid-enriched fraction and CPC subfractions), and 0.1 mg/mL (pure compounds). Each sample was injected at a volume of 2 μL. Data were analysed using the Bruker DataAnalysis 4.1 software.

^1^H and ^13^C (1D-NMR) and ^1^H/^1^H-COSY, ^1^H/^1^H-NOESY, ^1^H/^13^C-HSQC, and ^1^H/^13^C-HMBC (2D-NMR) spectra were recorded on an Agilent DD2 600 MHz spectrometer (Agilent, Santa Clara, CA, USA) at 26 °C in deuterated methanol (CD_3_OD). The recorded spectra were analysed with MestReNova version 15.0.0–34764 software and were referenced to the CD_3_OD solvent signals (^1^H: 3.310 ppm; and ^13^C: 49.000 ppm).

Circular dichroism (CD) spectra of compounds **5**, **17**, **18**, **19**, **21**, **22** and **24** were measured with a Jasco J-815 CD spectrometer (Jasco, Gros-Umstadt, Germany). The compounds were dissolved in CH_3_OH (0.1 mg/mL) and measurement using a 0.1 cm Suprasil Quartz cuvette (Hellma, Mullheim, Germany).

The three-dimensional molecular models were created using the Molecular Operating Environment (MOE; version 2018.0101) software. The molecular models were energy minimized with the MMFF94x force field, and then a Low-Mode Molecular Dynamics (LowModeMD) conformational search was performed using default settings of MOE. The conformers with the lowest energy were selected and energy minimized using the semi-empirical Austin Model 1 (AM1) Hamiltonian (MOPAC module of MOEs).

The simulated ECD curve for compound **24** (Appendix A) was obtained by Time-Dependent Density Functional Theory (TD-DFT) computations. Computational details are included in the caption of the Figure.

### 3.5. Spectral Data of the Isolated New Compounds

Cycloprotobuxoline-C *N*_20_-oxide (**2**): Yellow gum; +ESI-QqTOF-MS (*m*/*z*): 433.3773 [M + H]^+^ (calcd. for C_27_H_49_N_2_O_2_^+^: 433.3789), 217.1949 [M + 2H]^2+^ (calcd. for C_27_H_50_N_2_O_2_^2+^: 217.1931). For ^1^H- and ^13^C-NMR data (600/150MHz, CD_3_OD), see Table 3 and Table 4, respectively. For spectral data (LC/MS, ^1^H-NMR, ^13^C-NMR, and 2D-NMR (HSQC, COSY, HMBC, and NOESY), see Appendix A.

16α-Hydroxycycloprotobuxoline-C (**3**): White gum; +ESI-QqTOF-MS (*m*/*z*): 433.3840 [M + H]^+^ (calcd. for C_27_H_49_N_2_O_2_^+^: 433.3789), 217.2010 [M + 2H]^2+^ (calcd. for C_27_H_50_N_2_O_2_^2+^: 217.1931). For ^1^H- and ^13^C-NMR data (600/150MHz, CD_3_OD), see Table 3 and Table 4, respectively. For spectral data (LC/MS, ^1^H-NMR, ^13^C-NMR, and 2D-NMR (HSQC, COSY, HMBC, and NOESY), see Appendix A.

29-Trimethoxybenzoyloxy cycloprotobuxoline-C (**5**): Colorless gum; +ESI-QqTOF-MS (*m*/*z*): 627.4483 [M + H]^+^ (calcd. for C_37_H_59_N_2_O_6_^+^: 627.4368), 314.2334 [M + 2H]^2+^ (calcd. for C_37_H_60_N_2_O_6_^2+^+: 314.220). For ^1^H- and ^13^C-NMR data (600/150MHz, CD_3_OD), see Table 3 and Table 4, respectively. For spectral data (CD, UV, LC/MS, ^1^H-NMR, ^13^C-NMR, and 2D-NMR (HSQC, COSY, HMBC, and NOESY), see Appendix A.

*N*_3_-Demethylcycloprotobuxoline-C (**6**): Yellow gum; +ESI-QqTOF-MS (*m*/*z*): 403.3722 [M + H]^+^ (calcd. for C_26_H_47_N_2_O^+^: 403.3683), 202.1918 [M + 2H]^2+^ (calcd. for C_26_H_48_N_2_O^2+^: 202.1878). For ^1^H- and ^13^C-NMR data (600/150MHz, CD_3_OD), see Table 3 and Table 4, respectively. For spectral data (LC/MS, ^1^H-NMR, ^13^C-NMR, and 2D-NMR (HSQC, COSY, HMBC, and NOESY), see Appendix A.

16α-Hydroxy-*N*_3_-demethylcycloprotobuxoline-C (**7**): White gum; +ESI-QqTOF-MS (*m*/*z*): 419.3691 [M + H]^+^ (calcd. for C_26_H_47_N_2_O_2_^+^: 419.3632), 210.1910 [M + 2H]^2+^ (calcd. for C_26_H_48_N_2_O_2_^2+^: 210.1853). For ^1^H- and ^13^C-NMR data (600/150MHz, CD_3_OD), see Table 3 and Table 4, respectively. For spectral data (LC/MS, ^1^H-NMR, ^13^C-NMR, and 2D-NMR (HSQC, COSY, HMBC, and NOESY), see Appendix A.

Cycloprotobuxoline-D *N*_3_-*trans*- (**8a**) and cycloprotobuxoline-D *N*_3_-*cis* (**8b**) -formamide: White gum; +ESI-QqTOF-MS (*m*/*z*): 431.3678 (**8a**)/431.3671 (**8b**) [M + H]^+^ (calcd. for C_27_H_47_N_2_O_2_^+^: 431.3671), 216.1896 (**8a**)/216.1897 (**8b**) [M + 2H]^2+^ (calcd. for C_27_H_48_N_2_O_2_^2+^: 216.1853). For ^1^H- and ^13^C-NMR data (600/150MHz, CD_3_OD), see Table 3 and Table 4, respectively. For spectral data (LC/MS, ^1^H-NMR, ^13^C-NMR, and 2D-NMR (HSQC, COSY, HMBC, and NOESY), see Appendix A.

16a-Hydroxycycloprotobuxoline-C *N*_3_-trans-formamide (**9a**) and 16a-hydroxycycloprotobuxoline-C *N*_3_-*cis*-formamide (**9b**): White gum; +ESI-QqTOF-MS (*m*/*z*): 461.3808 (**9a**)/461.3775 (**9b**) [M + H]^+^ (calcd. for C_28_H_49_N_2_O_3_^+^: 461.3738), 231.1955 (**9a**)/231.1940 (**9b**) [M + 2H]^2+^ (calcd. for C_28_H_50_N_2_O_3_^2+^: 231.1905). For ^1^H- and ^13^C-NMR data (600/150MHz, CD_3_OD), see Table 3 and Table 4, respectively. For spectral data (LC/MS, ^1^H-NMR, ^13^C-NMR, and 2D-NMR (HSQC, COSY, HMBC, and NOESY), see Appendix A.

*N*_3_-Demethyl cyclonataminol (**11**): White gum; +ESI-QqTOF-MS (*m*/*z*): 431.3721 [M + H]^+^ (calcd. for C_27_H_47_N_2_O_2_^+^: 431.3632), 216.1937 [M + 2H]^2+^ (calcd. for C_27_H_48_N_2_O_2_^2+^: 216.1853). For ^1^H- and ^13^C-NMR data (600/150MHz, CD_3_OD), see Table 5 and Table 6, respectively. For spectral data (LC/MS, ^1^H-NMR, ^13^C-NMR, and 2D-NMR (HSQC, COSY, HMBC, and NOESY), see Appendix A.

Deoxycyclovirobuxeine-B (**12**): Yellow gum; +ESI-QqTOF-MS (*m*/*z*): 399.3785 [M + H]^+^ (calcd. for C_27_H_47_N_2_^+^: 399.3734), 200.1954 [M + 2H]^2+^ (calcd. for C_27_H_48_N_2_^2+^: 200.1903). For ^1^H- and ^13^C-NMR data (600/150MHz, CD_3_OD), see Table 5 and Table 6, respectively. For spectral data (LC/MS, ^1^H-NMR, ^13^C-NMR, and 2D-NMR (HSQC, COSY, HMBC, and NOESY), see Appendix A.

*N*_20_-Demethyl deoxycyclobuxoxazine A (**15**): Colorless gum; +ESI-QqTOF-MS (*m*/*z*): 415.3736 [M + H]^+^ (calcd. for C_27_H_47_N_2_O^+^: 415.3683), 208.1943 [M + 2H]^2+^ (calcd. for C_27_H_48_N_2_O^2+^: 208.1878). For ^1^H- and ^13^C-NMR data (600/150MHz, CD_3_OD), see Table 5 and Table 6, respectively. For spectral data (LC/MS, ^1^H-NMR, ^13^C-NMR, and 2D-NMR (HSQC, COSY, HMBC, and NOESY), see Appendix A.

Obtusibuxeine A (**16**): Yellow gum; +ESI-QqTOF-MS (*m*/*z*): 374.3115 [M + H]^+^ (calcd. for C_24_H_40_NO_2_^+^: 374.3054). For ^1^H- and ^13^C-NMR data (600/150MHz, CD_3_OD), see Table 5 and Table 6, respectively. For spectral data (LC/MS, ^1^H-NMR, ^13^C-NMR, and 2D-NMR (HSQC, COSY, HMBC, and NOESY), see Appendix A.

*O*^10^-Obtusifuranamine-A (**17**): Yellow gum; +ESI-QqTOF-MS (*m*/*z*): 593.3679 [M + H]^+^ (calcd. for C_35_H_49_N_2_O_6_^+^: 593.3585), 297.1921 [M + 2H]^2+^ (calcd. for C_35_H_50_N_2_O_6_^2+^: 297.1829). For ^1^H- and ^13^C-NMR data (600/150MHz, CD_3_OD), see Table 5 and Table 6, respectively. For spectral data (CD, UV, LC/MS, ^1^H-NMR, ^13^C-NMR, and 2D-NMR (HSQC, COSY, HMBC, and NOESY), see Appendix A.

*O*^10^-Obtusifuranamine-B (**18**): Yellow gum; +ESI-QqTOF-MS (*m*/*z*): 655.3817 [M + H]^+^ (calcd. for C_40_H_51_N_2_O_6_^+^: 655.3742), 328.1977 [M + 2H]^2+^, (calcd. for C_40_H_52_N_2_O_6_^2+^: 328.1907). For ^1^H- and ^13^C-NMR data (600/150MHz, CD_3_OD), Table 5 and Table 6, respectively. For spectral data (CD, UV, LC/MS, ^1^H-NMR, ^13^C-NMR, and 2D-NMR (HSQC, COSY, HMBC, and NOESY), see Appendix A.

16-Deoxy-*O*^10^-obtusifuranamine-B (**19**): Colorless gum; +ESI-QqTOF-MS (*m*/*z*): 639.3602 [M + H]^+^ (calcd. for C_40_H_51_N_2_O_5_^+^: 639.3792), 320.1993 [M + 2H]^2+^ (calcd. for C_40_H_52_N_2_O_5_^2+:^ 320.1933). For ^1^H- and ^13^C-NMR data (600/150MHz, CD_3_OD), see Table 5 and Table 6, respectively. For spectral data (CD, UV, LC/MS, ^1^H-NMR, ^13^C-NMR, and 2D-NMR (HSQC, COSY, HMBC, and NOESY), see Appendix A.

Obtusiepoxamine-A (**21**): Yellow gum; +ESI-QqTOF-MS (*m*/*z*): 577.3728 [M + H]^+^ (calcd. for C_35_H_49_N_2_O_5_^+^: 577.3636), 289.1940 [M + 2H]^2+^ (calcd. for C_35_H_50_N_2_O_5_^2+^: 289.1855). For ^1^H- and ^13^C-NMR data (600/150MHz, CD_3_OD), see Table 5 and Table 6, respectively. For spectral data (CD, UV, LC/MS, ^1^H-NMR, ^13^C-NMR, and 2D-NMR (HSQC, COSY, HMBC, and NOESY), see Appendix A.

Obtusidienolamine-A (**22**): Colorless gum; Yellow gum; +ESI-QqTOF-MS (*m*/*z*): 579.3887 [M + H]^+^ (calcd. for C_35_H_51_N_2_O_5_^+^: 579.3792), 290.2016 [M + 2H]^2+^ (calcd. for C_35_H_52_N_2_O_5_^2+^: 290.1933). For ^1^H- and ^13^C-NMR data (600/150MHz, CD_3_OD), see Table 5 and Table 6, respectively. For spectral data (CD, UV, LC/MS, ^1^H-NMR, ^13^C-NMR, and 2D-NMR (HSQC, COSY, HMBC, and NOESY), see Appendix A.

Deoxyobtusidienolamine-A (**23**): Colorless gum; +ESI-QqTOF-MS (*m*/*z*): 563.3801 [M + H]^+^ (calcd. for C_35_H_51_N_2_O_4_^+^: 563.3843), 282.1999 [M + 2H]^2+^ (calcd. for C_35_H_52_N_2_O_4_^2+^: 282.1958). For ^1^H- and ^13^C-NMR data (600/150MHz, CD_3_OD), see Table 5 and Table 6, respectively. For spectral data (LC/MS, ^1^H-NMR, ^13^C-NMR, and 2D-NMR (HSQC, COSY, HMBC, and NOESY), see Appendix A.

Obtusiaminocyclin (**24**): Yellow gum; +ESI-QqTOF-MS (*m*/*z*): 368.2660 [M + H]^+^ (calcd. for C_24_H_34_NO_2_^+^: 368.2584). For ^1^H- and ^13^C-NMR data (600/150MHz, CD_3_OD), see Table 7. For spectral data (ECD, UV, LC/MS, ^1^H-NMR, ^13^C-NMR, and 2D-NMR (HSQC, COSY, HMBC, and NOESY), see Appendix A.

### 3.6. In Vitro Bioassays

In vitro biological activity against *Trypanosoma brucei rhodesiense* (bloodstream trypomastigotes, STIB 900 strain), *Plasmodium falciparum* (intraerythrocytic form, NF54 strain) and cytotoxicity assay against rat skeletal myoblasts (L6 cell line) were carried out at the Swiss Tropical and Public Health Institute (Swiss TPH, Allschwil, Switzerland) in accordance with the established standard procedures, in exactly the same manner and with the same cell lines as described recently [14].

## 4. Conclusions

In the present study, a detailed phytochemical investigation of the dichloromethane extract from the leaves of *Buxus obtusifolia* was conducted for the first time. A total of 24 aminosteroid alkaloids were successfully isolated and characterized, confirming the plant’s richness as a source of antiprotozoal compounds. Among the isolated natural products, 18 were previously undescribed, including two compounds with novel carbocyclic skeletons (**16** and **24**), which could all be elucidated by their NMR and LC/MS data. Of particular note is compound **24**, which possesses a unique hexacyclic skeleton featuring a cyclic amine alongside an unprecedented position of the cyclopropane moiety involving carbons C-9, C-19 and C-11. This structural arrangement represents a significant novelty amongst the isolated compounds. Interestingly, approximately 68% of the 9-19-cyclo-5α-pregnanes identified in our study (compounds **1**–**11**) possess a hydroxy group at C-2, which is a rare structural feature among *Buxus* alkaloids. This characteristic may therefore serve as a potential chemotaxonomic marker for *B. obtusifolia*.

The alkaloid fraction, several CPC subfractions and a number of isolated alkaloids displayed prominent antiprotozoal activities, with compounds **12** and **5** showing the highest activities (IC_50_ values of <1.0 µmol/L) against *Trypanosoma brucei rhodesiense* and *Plasmodium falciparum*, respectively. This work further complements the ongoing studies of our group on the structural diversity of antiprotozoal aminosteroid alkaloids and contributes valuable molecular candidates that may serve as promising lead candidates for the development of therapies against Malaria and sleeping sickness. Three-dimensional quantitative structure–activity relationship (3D–QSAR) analyses of all aminosteroids obtained from *B. obtusifolia* in comparison with those previously reported from *P. terminalis*, *B. sempervirens* and *H. africana* by our laboratory are currently underway. Additionally, detailed mechanistic or molecular target studies will be interesting subjects for future studies. In order to investigate the dynamics of alkaloid accumulation in *B. obtusifolia*, particularly to determine the optimal harvesting period for isolating the bioactive alkaloids, a study on seasonal and climate-related variation is currently in progress.

## Data Availability

The original data of this study is detailed in the article and Appendix A. The raw data supporting the conclusions of this article will be made available by the authors on request.

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
