# Peer review of "Antiprotozoal Aminosteroid Alkaloids from Buxus obtusifolia (Mildbr.) Hutch."

_molecules, 2025, doi:10.3390/molecules30234558_

Round 1
Reviewer 1 Report
Comments and Suggestions for Authors
This paper presents the phytochemical and biological investigation of Buxus obtusifolia, a shrub native to Kenya and Tanzania belonging to the Buxaceae family. The study aimed to isolate/characterize aminosteroid alkaloids and to evaluate their antiprotozoal activity against Trypanosoma brucei rhodesiense (Tbr) and Plasmodium falciparum (Pf). A total of 24 alkaloids were isolated, 18 of which were new natural products. Two of these (compounds 1 and 4) were identified as natural constituents for the first time. Identification was performed using UHPLC/ESI-QqTOF-MS and extensive 1D and 2D NMR analyses. Structural assignments were carefully justified through comparisons of chemical shifts, NOE correlations, and biogenetic reasoning. Notably, the elucidation of compound 24, which features a novel carbocyclic core with a cyclopropane bridge, represents a significant structural discovery. Antiprotozoal and cytotoxicity tests were carried out according to standard WHO/TDR protocols at the Swiss Tropical and Public Health Institute. The inclusion of selectivity indices lends credibility to the biological evaluation. The structure–activity relationship (SAR) suggests that N-methylation at C-3 and C-20 and the presence of benzoate or substituted benzoate groups are key to potency — a finding consistent with related studies of Buxus and Pachysandra.
Based on the obtained results, this referee suggests the acceptance of this paper in the present form, but some minor points could be answered, as follows:
• Lack of mechanistic insight: Although SARs are proposed, no mechanistic or molecular target analysis (e.g. enzyme inhibition or membrane disruption) has been conducted.
• Statistical robustness: Some activity values are based on two replicates; increasing the number of replicates would enhance reliability.
Author Response
Reviewer 1
This paper presents the phytochemical and biological investigation of Buxus obtusifolia, a shrub native to Kenya and Tanzania belonging to the Buxaceae family. The study aimed to isolate/characterize aminosteroid alkaloids and to evaluate their antiprotozoal activity against Trypanosoma brucei rhodesiense (Tbr) and Plasmodium falciparum (Pf). A total of 24 alkaloids were isolated, 18 of which were new natural products. Two of these (compounds 1 and 4) were identified as natural constituents for the first time. Identification was performed using UHPLC/ESI-QqTOF-MS and extensive 1D and 2D NMR analyses. Structural assignments were carefully justified through comparisons of chemical shifts, NOE correlations, and biogenetic reasoning. Notably, the elucidation of compound 24, which features a novel carbocyclic core with a cyclopropane bridge, represents a significant structural discovery. Antiprotozoal and cytotoxicity tests were carried out according to standard WHO/TDR protocols at the Swiss Tropical and Public Health Institute. The inclusion of selectivity indices lends credibility to the biological evaluation. The structure–activity relationship (SAR) suggests that N-methylation at C-3 and C-20 and the presence of benzoate or substituted benzoate groups are key to potency — a finding consistent with related studies of Buxus and Pachysandra.
Based on the obtained results, this referee suggests the acceptance of this paper in the present form, but some minor points could be answered, as follows:
- Lack of mechanistic insight: Although SARs are proposed, no mechanistic or molecular target analysis (e.g. enzyme inhibition or membrane disruption) has been conducted.
It was the aim of the present study to find new biologically active aminosteroids. Mechanistic or molecular target analysis will be interesting subjects for further studies. A corresponding statement has been added to the conclusions.
- Statistical robustness: Some activity values are based on two replicates; increasing the number of replicates would enhance reliability.
It is common practice in the established protocols at SwissTPH to report such data based on two independent measurements if the absolute value of the deviation is not greater than the dilution step, which is 3-fold in our bioassays. (Example: In case of compound 1, the mean of 6.7 µM is derived from measured values of 10.0 and 3.4; since the absolute deviation = 3.3 is not greater than the upper value 10.0 divided by 3, no additional determination was performed, even though this is a borderline case). However, since it is better in case of only two measurements to report the absolute deviations of the measured data from their means and not the standard deviations, we adjusted the deviation values in tables 1, 2 and 8.
We thank the reviewer for the thorough assessment and the time and effort spent to help us improve our manuscript.
Reviewer 2 Report
Comments and Suggestions for Authors
This manuscript systematically reports the isolation and identification of 24 aminosteroid alkaloids from the Kenyan endemic plant Buxus obtusifolia, along with an evaluation of their in vitro antiprotozoal activity against the parasites responsible for Human African Trypanosomiasis and Malaria. The study possesses notable strengths, including the discovery of 18 new compounds, among which two feature novel skeletons (Obtusibuxeine A (16) and Obtusiaminocyclin (24)), and the identification of three compounds (5, 6, 12) with nanomolar-level potency. A preliminary discussion on the structure-activity relationships (SAR) is also provided. The supporting data reflect a highly professional and rigorous approach. Overall, this is an excellent, high-quality paper well-suited for publication in Molecules.
Prior to publication, the following minor points deserve attention:
- Highlighting Novel Structures:The discovery of new compounds, particularly those with novel skeletons, is a key merit of this work. To emphasize this, the authors could consider presenting the most significant structure, such as compound 24, earlier in the "Results and Discussion" section. Furthermore, as the structural elucidation, including stereochemistry, primarily relies on biosynthetic reasoning (with no single-crystal X-ray data for absolute configuration), it is highly recommended to strengthen the assignment for the flagship compound 24. Performing a comparison between its experimental ECD spectrum and time-dependent Density Functional Theory (TD-DFT) calculated ECD curves would provide more definitive evidence for its absolute configuration. Additionally, including the key 2D NMR correlations (COSY, HSQC, HMBC, and NOE) for 24 in the main text instead of SI as a representative example would effectively illustrate the detailed NMR-based structural elucidation process.
- Missing Spectroscopic Data:The physicochemical data for the compounds lack specific optical rotations and ECD spectra. Including this data is crucial, as it allows for comparison with known compounds from the literature and facilitates future absolute configurational assignments.
- Biosynthetic Considerations:The authors frequently invoke biosynthetic arguments to support stereochemical assignments. Therefore, it would be beneficial to include a brief discussion or a schematic outlining the proposed biosynthetic pathway for these alkaloids, especially for the new scaffold compounds. This would provide valuable context and inspiration for future synthetic studies.
- SAR Analysis Presentation:The SAR discussion could be enhanced for clarity. It is recommended to include a diagram of the core alkaloid skeleton, annotating the key functional groups and substitution sites that influence activity. If feasible, a pharmacophore model analysis or a more systematic comparison of the impact of different substituents (e.g., C-2 OH, C-16 OH, C-29 ester) on activity would further deepen the SAR insights.
- Figure and Table Presentation:In Figure 1, the alignment of some R-group labels could be improved for better visual clarity. In Table 8, for consistency and easier comparison, it is suggested to use only micromolar (µM) units for the ICâ‚…â‚€ values, removing the concurrent use of µg/mL.
Author Response
Reviewer 2
This manuscript systematically reports the isolation and identification of 24 aminosteroid alkaloids from the Kenyan endemic plant Buxus obtusifolia, along with an evaluation of their in vitro antiprotozoal activity against the parasites responsible for Human African Trypanosomiasis and Malaria. The study possesses notable strengths, including the discovery of 18 new compounds, among which two feature novel skeletons (Obtusibuxeine A (16) and Obtusiaminocyclin (24)), and the identification of three compounds (5, 6, 12) with nanomolar-level potency. A preliminary discussion on the structure-activity relationships (SAR) is also provided. The supporting data reflect a highly professional and rigorous approach. Overall, this is an excellent, high-quality paper well-suited for publication in Molecules.
Prior to publication, the following minor points deserve attention:
- Highlighting Novel Structures:The discovery of new compounds, particularly those with novel skeletons, is a key merit of this work. To emphasize this, the authors could consider presenting the most significant structure, such as compound 24, earlier in the "Results and Discussion" section. Furthermore, as the structural elucidation, including stereochemistry, primarily relies on biosynthetic reasoning (with no single-crystal X-ray data for absolute configuration), it is highly recommended to strengthen the assignment for the flagship compound 24. Performing a comparison between its experimental ECD spectrum and time-dependent Density Functional Theory (TD-DFT) calculated ECD curves would provide more definitive evidence for its absolute configuration. Additionally, including the key 2D NMR correlations (COSY, HSQC, HMBC, and NOE) for 24 in the main text instead of SI as a representative example would effectively illustrate the detailed NMR-based structural elucidation process.
The position of compound 24 as the last compound to be described and discussed was chosen very deliberately, in order to highlight its special structure on the background of all the other structures with molecular scaffolds more common for a Buxus species. We would therefore rather leave this unchanged. However, we agree with the reviewer to include its key NMR correlations in the main manuscript; the information has been added to Figure 5.
As suggested, we have also recorded the ECD spectrum and performed a TD-DFT simulation which gave an almost perfect match. This comparison has been added as Figure S210.
- Missing Spectroscopic Data:The physicochemical data for the compounds lack specific optical rotations and ECD spectra. Including this data is crucial, as it allows for comparison with known compounds from the literature and facilitates future absolute configurational assignments.
The CD data have been added in all cases where this was feasible (compounds 5 (Figure S22), 17 (Figure S133), 18 (Figure S145),19 (Figure S158), 21 (Figure S171) and 22 (Figure S184)). Optical rotation values were not obtained due to the low amounts of the compounds which would not yield very reliable measurements. We hope the reviewer will be satisfied with the CD information wherever it could be obtained.
- Biosynthetic Considerations:The authors frequently invoke biosynthetic arguments to support stereochemical assignments. Therefore, it would be beneficial to include a brief discussion or a schematic outlining the proposed biosynthetic pathway for these alkaloids, especially for the new scaffold compounds. This would provide valuable context and inspiration for future synthetic studies.
Our stereochemical assignments and arguments are based on the common biosynthetic pathway of cycloartane triterpenoids which leads to the relative and absolute configurations analogous to the mother triterpene, cycloartenol. Any hypothetical considerations on specific (enzymatic) steps going beyond this and leading to the new scaffolds would be mere speculation which we would rather not add to our publication.
- SAR Analysis Presentation: The SAR discussion could be enhanced for clarity. It is recommended to include a diagram of the core alkaloid skeleton, annotating the key functional groups and substitution sites that influence activity. If feasible, a pharmacophore model analysis or a more systematic comparison of the impact of different substituents (e.g., C-2 OH, C-16 OH, C-29 ester) on activity would further deepen the SAR insights.
A thorough investigation of (quantitative) structure-activity-relationships of the compounds of this study along with many related compounds previously isolated and tested by our group is in progress and will be subject to a separate communication. We therefore do not wish to add any further SAR insights limited to the compounds of the present study. We hope the reviewer and editor can accept this.
- Figure and Table Presentation:In Figure 1, the alignment of some R-group labels could be improved for better visual clarity. In Table 8, for consistency and easier comparison, it is suggested to use only micromolar (µM) units for the ICâ‚…â‚€ values, removing the concurrent use of µg/mL.
We have aligned the R-group labels in Figure 1 as suggested by the reviewer.
With respect to Table 8, the reviewer is right, we have removed the µg/mL data and transformed the deviations to molar scale as well.
We thank the reviewer for the thorough assessment and the time and effort spent to help us improve our manuscript.